# A Polyamine Oxidase from *Selaginella lepidophylla* (SelPAO5) can Replace AtPAO5 in *Arabidopsis* through Converting Thermospermine to Norspermidine instead to Spermidine

**DOI:** 10.3390/plants8040099

**Published:** 2019-04-15

**Authors:** G. H. M. Sagor, Tomonobu Kusano, Thomas Berberich

**Affiliations:** 1Department of Genetics and Plant Breeding, Faculty of Agriculture, Bangladesh Agricultural University, Mymensingh 2202, Bangladesh; sagorgpb@gmail.com; 2Graduate School of Life Sciences, Tohoku University, 2-1-1 Katahira, Aoba, Sendai 980-8577, Japan; kusano@ige.tohoku.ac.jp; 3Laboratory Center, Senckenberg Biodiversity and Climate Research Center, Georg-Voigt-Str. 14-16, D-60325 Frankfurt am Main, Germany

**Keywords:** polyamine oxidase, norspermidine, thermospermine, *Selaginella lepidophylla*, *Arabidopsis thaliana* mutant

## Abstract

Of the five polyamine oxidases in *Arabidopsis thaliana*, AtPAO5 has a substrate preference for the tetraamine thermospermine (T-Spm) which is converted to triamine spermidine (Spd) in a back-conversion reaction in vitro. A homologue of AtPAO5 from the lycophyte *Selaginella lepidophylla* (SelPAO5) back-converts T-Spm to the uncommon polyamine norspermidine (NorSpd) instead of Spd. An *Atpao5* loss-of-function mutant shows a strong reduced growth phenotype when growing on a T-Spm containing medium. When SelPAO5 was expressed in the *Atpao5* mutant, T-Spm level decreased to almost normal values of wild type plants, and NorSpd was produced. Furthermore the reduced growth phenotype was cured by the expression of *SelPAO5*. Thus, a NorSpd synthesis pathway by PAO reaction and T-Spm as substrate was demonstrated in planta and the assumption that a balanced T-Spm homeostasis is needed for normal growth was strengthened.

## 1. Introduction

Polyamines (PAs) are aliphatic compounds derived from amino acids with low molecular masses that are ubiquitously present in all living organisms [1,2]. Plants mainly contain the diamine putrescine (Put), the triamine spermidine (Spd), and the two tetraamines spermine (Spm) and thermospermine (T-Spm) [3,4,5,6,7], an isomer of Spm that was first discovered in thermophilic bacteria [8]. They are implicated in regulating various developmental processes such as embryogenesis, cell division, organogenesis, flowering, and senescence, as well as responses to abiotic and biotic stresses [9,10,11,12,13,14]. The biosynthesis of the polyamines Put, Spd, Spm, and T-Spm in plants is well elucidated [15]. “Lower” or non-vascular plants, such as bryophytes, mosses, and some eukaryotic algae, contain norspermidine (NorSpd) and norspermine (NorSpm) [16,17,18]. The biosynthesis of those uncommon PAs starts with 1,3-diaminopropane (DAP), which is produced by the metabolism of Spd and Spm through the action of terminal catabolism-type polyamine oxidase (PAO) [19]. The aminopropyl residue derived from decarboxylated S-adenosylmethionine is transferred to DAP by a putative aminopropyltransferase (APT) with relaxed substrate specificity, resulting in NorSpd, and subsequently, a second APT action converts NorSpd to NorSpm [19]. However, the occurrence of NorSpd and NorSpm has also been reported in alfalfa [20] and maize [21]. Catabolism of PAs in plants is executed by two kind of oxidases, copper-dependent amine oxidase (CuAO) and flavin-containing polyamine oxidase (PAO). PAO are reported to act in two different pathways, a terminal catabolic pathway and a back-conversion pathway [22]. The first characterized plant PAOs of maize and barley catalyze the terminal catabolic reactions [23,24,25,26,27]. They oxidize the carbon at the endo-side of the N4-nitrogen of Spm and Spd, producing N-(3-aminopropyl)-4-aminobutanal and 4-aminobutanal, respectively, and concomitantly 1,3-diaminopropane and H_2_O_2_ in both reactions [28,29]. A back-conversion reaction was first shown for *Arabidopsis thaliana* PAO1 that produces Spd from Spm and NorSpd from NorSpm in vitro [30]. The *Arabidopsis thaliana* gene family of PAO comprises five members named *AtPAO1*–*AtPAO5* with well characterized gene products that all function in the back-conversion of tetraamines to triamines and/or triamines to diamines, albeit with different substrate specificities [22]. AtPAO1 localizes in the cytoplasm and oxidizes Spm, T-Spm, and NorSpm, but not Spd [30], while AtPAO2, AtPAO3, and AtPAO4 localize in peroxisomes [31,32]. AtPAO2 and AtPAO3 convert Spm to Put via Spd, whereas AtPAO4 produces less Put from Spm, which is explained by the very low affinity for Spd [33]. AtPAO5 localizes in the cytoplasm and shows a preference to convert T-Spm (or Spm) to Spd [34]. *Arabidopsis pao5* mutants contain 2-fold higher T-Spm levels exhibit aerial tissue growth retardation and growth inhibition of stems and leaves at an early stage of development after external T-Spm application [4]. These findings are in accord with observations made in *Arabidopsis* plants with mutated *acaulis5* (*ACL5*) gene encoding T-Spm synthase. In this mutant (*acl5*), T-Spm content is reduced producing a dwarf phenotype with over-proliferated xylem vessels, suggesting a role of T-Spm in xylem differentiation [4,5]. Taken together, a fine-tuned T-Spm homeostasis secured by regulation of T-Spm synthase (Acaulis5) and T-Spm oxidase (AtPAO5) activities is necessary for proper xylem development and growth. A PAO from the lycophyte *Selaginella lepidophylla* (SelPAO5) with the highest sequence identity to AtPAO5 was shown to prefer T-Spm and Spm as substrates like the *Arabidopsis* homologue, but instead back-converts T-Spm to NorSpd not to Spd [35]. Here, for further characterization, we used the SelPAO5 encoding cDNA to complement the *Arabidopsis Atpao5* mutant.

## 2. Results

### 2.1. Phylogenetic Classification of SelPAO5 and Cellular Localization

Recombinant proteins of *Arabidopsis* AtPAO5 and rice OsPAO1 both prefer Spm and T-Spm as substrates and back-convert it to Spd in vitro [34,36]. These two PAOs are considered to convert T-Spm to Spd in plants. Phylogenetic relationship of PAOs identified in the genome of *Selaginella moellendorffii* [37] and SelPAO5 of *S. lepidophylla* to PAOs of *Arabidopsis* and rice is shown in Figure 1. PAO6 and PAO7 of *S. moellendorffii* (SmPAO6 and SmPAO7) are members of the clade III plant PAOs that comprise AtPAO5 and OsPAO1. SelPAO5 of *S. lepidophylla* belongs to this clade and is the homologue to SmPAO6 and SmPAO7.

### 2.2. SelPAO5 Complementation of Atpao5-2 Mutant Rescues T-Spm-Induced Growth Inhibition

For complementing the mutated *AtPAO5* gene in *Atpao5-2* with *SelPAO5*, the cDNA of *SelPAO5* was introduced into *Atpao5-2*. *Atpao5-2* plants and *Atpao5-2* plants transformed with empty vector displayed reduced growth on T-Spm containing medium compared to wild type plants (Figure 2A), while *Atpao5-2* plants transformed with the vector containing *SelPAO5* cDNA did not. Instead, they looked similar to wild type plants (Figure 2A). For quantification of plant growth, the average fresh weight of ten seedlings each was compared (Figure 2B). While wild type and *Atpao5-2* plants expressing *SelPAO5* had an average weight of about 75 mg; the growth reduced *Atpao5-2* plants and *Atpao5-2* plants transformed with the empty vector had an average weight of 50 mg (Figure 2B). Expression of SelPAO5 was confirmed by RT-PCR with RNA samples of wild type *Arabidopsis* plants, *Atpao5-2* plants, and *Atpao5-2* plants transformed with the empty vector or the vector containing *SelPAO5* cDNA (lines S5#5, S5#11 and S5#13), respectively (Appendix AA). *AtPAO5* expression could only be detected in wild type plants but not in the *Atpao5-2* mutants. *SelPAO5* expression was confirmed in three independent *Atpao5-2* lines that were transformed with the *SelPAO5* cDNA-containing vector but not in plants that have been transformed with the empty vector only. While T-Spm had a negative effect on growth of *Atpao5-2* plants, and *Atpao5-2* plants transformed with the empty vector (Appendix AB), other polyamines, Put, Spd, and Spm, respectively, did not have such an effect (Appendix AC). In all the three lines of *Atpao5-2* expressing *SelPAO5*, no growth inhibition could be seen on T-Spm containing medium.

### 2.3. SelPAO Produces NorSpd in Arabidopsis Plants

Polyamine patterns in the *Atpao5-2* mutant expressing *SelPAO5* were compared to that of wild type *Arabidopsis* (Col-0) by HPLC analysis (Figure 3). In Col-0 plants, the major plant PAs Put, Spd, T-Spm, and Spm were detected but not NorSpd. The *Atpao5-2* mutant expressing *SelPAO5* (*pao5-2/SelPAO* OX) contained NorSpd in addition to the four other PAs. Quantification of PAs revealed that three lines of *Atpao5-2* mutant expressing *SelPAO5* (S5#5, S5#11, and S5#13) contained more Put (10–12 nmol/gFW) than wild type *Arabidopsis* plants, *Atpao5-2* plants, and *Atpao5-2* plants transformed with the empty vector (7–8 nmol/gFW, Figure 4A). Spd levels were similar (~50 nmol/gFW) among these plants (Figure 4B). T-Spm content in *Atpao5-2* plants and *Atpao5-2* plants transformed with the empty vector were higher (~8 nmol/gFW) than in wild type and *Atpao5-2* plants expressing *SelPAO5* (~5 nmol/gFW). The Spm content did not vary much (10–15 nmol/gFW) within the plants tested (Figure 4D). NorSpd was only detected in *Atpao5-2* plants expressing *SelPAO5* (5-9 nmol/gFW, Figure 4E). The data show that SelPAO5 produces NorSpd when expressed in the *Atpao5-2* background. Furthermore, since the level of T-Spm drops in the *SelPAO5* expressing *Atpao5-2* mutant compared to untransformed *Atpao5-2* and *Atpao5-2* transformed with the empty vector, and the Spm contents stay almost same, it can be assumed that SelPAO5 converts T-Spm to NorSpd in planta.

## 3. Discussion

A recent phylogenetic analysis using a plant PAO protein sequence database identified four subfamilies: three subfamilies comprising PAOs with back conversion activity named PAO back conversion 1–3 (PAObc1, PAObc2, PAObc3), and one subfamily formed by terminal catabolism PAOs (subfamily PAOtc) [40]. PAObc1 was present on every lineage in the survey, pointing out important roles of back conversion-type PAOs in plants. PAObc2 was exclusively present in vascular plants, supporting the idea that T-Spm oxidase activity plays an important role in the development of the vascular system [34,40]. *Arabidopsis* AtPAO5 and rice OsPAO1 belong to this subfamily. Based on phylogenetic relationship, polyamine oxidase SelPAO5 of *Selaginella lepidophylla* is an orthologue of *Arabidopsis* AtPAO5 and rice OsPAO1 which both convert Spm and T-Spm to Spd in a back-conversion reaction. Therefore, it was expected that SelPAO5 produces Spd when using T-Spm as a substrate. However, in a previous work we could show that the recombinant SelPAO5 protein produces NorSpd in vitro [35]. To further characterize SelPAO5, we wanted to answer the questions i) does SelPAO5 convert Spm and/or T-Spm to NorSpd in vivo, and ii) can SelPAO5 replace AtPAO5 function and cure the reduced growth phenotype of *Atpao5-2* mutant? In the *Atpao5-2* mutant, T-Spm levels were increased, and plants showed a reduced growth phenotype [34]. A reduced growth phenotype was also observed in the *Arabidopsis Acaulis5* mutant (*acl5*) lacking T-Spm synthase activity and thus had decreased T-Spm levels [4,5,41]. Therefore, it is assumed that deviation from normal T-Spm levels, both an increase and decrease, cause reduced growth of plants [34]. A balanced homeostasis of T-Spm is necessary for normal growth. When SelPAO5 was expressed in the *Arabidopsis Atpao5-2* mutant, the T-Spm content decreased to almost normal levels of wild type plants while Spm levels did not decrease. NorSpd was only detected in the *Atpao5-2* mutant that expressed *SelPAO5*. In total, these results suggest that SelPAO5 uses T-Spm as substrate and converts it to NorSpd in a back-conversion reaction when expressed in *Arabidopsis*. Reduction of T-Spm content to almost wild type levels by SelPAO5 action also cured the growth retardation effect that is caused by increased T-Spm levels and enables normal development. Whether Spd or NorSpd is produced by the T-Spm specific PAO activity does not make a difference concerning the effect of T-Spm homeostasis. The presence of NorSpd does not seem to disturb development of *Arabidopsis*, although it is usually not detectable in this plant. NorSpd is an unusual triamine in eukaryotes, which is present in lower, single-celled eukaryotes including *Euglena*, cryptophytes, diatoms, and also in *Chlamydomonas* and *Volvox* [18,42,43], but also in Bryophytes [16], in the leguminous plant *Medicago sativa* (alfalfa) [20], and in maize [21]. A NorSpd synthesis pathway like in the Gram-negative bacterium *Vibrio cholerea* is not found in eukaryotes [7], and NorSpd synthesis in alfalfa from the precursor DAP (1,3-diaminopropane), which is a co-product of Spd oxidation by PAO, could not be demonstrated [44]. The production of NorSpd by SelPAO5 using T-Spm as a substrate is a demonstration of a NorSpd synthesis pathway in plants. The idea that T-Spm back-conversion by a PAO results in NorSpd in plants is backed by the finding that presence of homologues of the *Arabidopsis ACL5*-encoded T-Spm synthase in genomes correlates with the presence of NorSpd in the organism [7]. In the unicellular green alga *Chlamydomonas reinhardtii*, NorSpd stimulated cell division [45]. The role that NorSpd could play in higher plants is yet unknown. Further work should be done to follow how NorSpd is further metabolized in *Arabidopsis* and what kind of effect it has by making use of the *Atpao5-2* mutant expressing *SelPAO5*.

## 4. Materials and Methods

### 4.1. Plant Materials and Growth Conditions

*Arabidopsis thaliana* wild-type (WT) plants [accession Columbia-0 (Col-0)] and the T-DNA inserted *Atpao5-2* line (SALK_053110) [35] were used in this work. All seeds were surface sterilized by wetting with 70% ethanol for 1 min and subsequent treatment in a solution of 1% sodium hypochloride and 0.1% Tween-20 for 15 min. After extensive washing with sterile distilled water, sterilized seeds were placed onto vermiculite or on 1/2 Murashige and Skoog medium-1.5% agar plates (pH 5.6) containing 1% sucrose. Agar plates were kept upright under the angle of 75° to ground to allow plant growth on the agar surface by gravity. Growth conditions were 22 °C with a 14 h light/10 h dark photocycle.

### 4.2. Determination of Plant Fresh Weight

Seedlings grown for 24 days on agar surface containing 5 µM T-Spm where carefully picked with forceps and immediately weighed on a precision scale. Statistical analysis was done using MS-Excel software.

### 4.3. Chemicals

Put, Spd, and Spm were purchased from Nacalai-Tesque Ltd. (Kyoto, Japan). T-Spm and Nor-Spd was chemically synthesized [46]. All other analytical grade chemicals were obtained from Sigma-Aldrich Corp. (St. Louis, MO, USA), Wako Pure Chemical Industries Ltd. (Osaka, Japan), and Nacalai-Tesque Ltd.

### 4.4. Generation of Arabidopsis pao5 T-DNA Insertion Mutant Transgenic Lines Expressing SelPAO5 ORF

The fragment encompassing the coding region of the *SelPAO5* cDNA was amplified by PCR with the primer pair listed in Appendix A. It was digested with *Xba*I and *Sac*I and subcloned into the corresponding sites of the pPZP2Ha3(+) vector [39], yielding pPZP2Ha3(+)-SelPAO5. This plasmid was introduced into *Agrobacterium tumefaciens* strain GV3101, and the *Agrobacterium* transformant then introduced into *pao5-2* plants using the floral dip method [47]. The resulting seeds were selected on MS agar medium containing 25 mg/mL hygromycin (hyg) and 50 mg/mL carbenicillin. T_2_ seeds, obtained from self-fertilization of primary transformants, were surface-sterilized and grown on hyg-containing plates. Seedlings showing a 3:1 (resistant: sensitive) segregation ratio were selected to produce homozygous (hyg^R^/hyg^R^) T_3_ lines that were used for further study.

### 4.5. RT-PCR Analysis

Total RNA was extracted from whole aerial parts of two-week-old *Arabidopsis* seedlings using Sepasol-RNA I Super (Nacalai-Tesque, Kyoto, Japan). First-strand cDNA was synthesized with ReverTra Ace (Toyobo Co. Ltd., Osaka, Japan) and oligo-dT primers. Quantitative real-time RT-PCR was performed in triplicate using Fast-Start Universal SYBR Green Master (ROX; Roche Molecular Systems, Indianapolis, IN, USA) on a StepOne real-time PCR system (Thermo Fisher Scientific, Waltham, MA, USA) using the above cDNA and the primers listed in Appendix A. Constitutively expressed *AtActin* (accession number, NC_008396.2) was used as an internal control for the analysis to which the amount of target mRNA was normalized.

### 4.6. PA Analysis by High-Performance Liquid Chromatography (HPLC)

PA analysis was performed as described previously [5]. The benzoylated PAs were analyzed with a programmable Hewlett Packard series 1200 liquid chromatograph using a reverse-phase column (4.6 × 250 mm, TSK-GEL ODS-80Ts, TOSOH, Tokyo, Japan) and detected at 254 nm. One cycle of the run consisted of a total of 60 min at a flow rate of 1 mL/min at 30 °C; i.e., 42% acetonitrile for 25 min for PA separation, increased up to 100% acetonitrile during 3 min, then 100% acetonitrile for 20 min for washing, decreased down to 42% acetonitrile during 3 min, and finally 42% acetonitrile for 9 min. Statistical analysis was done using MS-Excel software.

## Figures and Tables

**Figure 1 plants-08-00099-f001:**
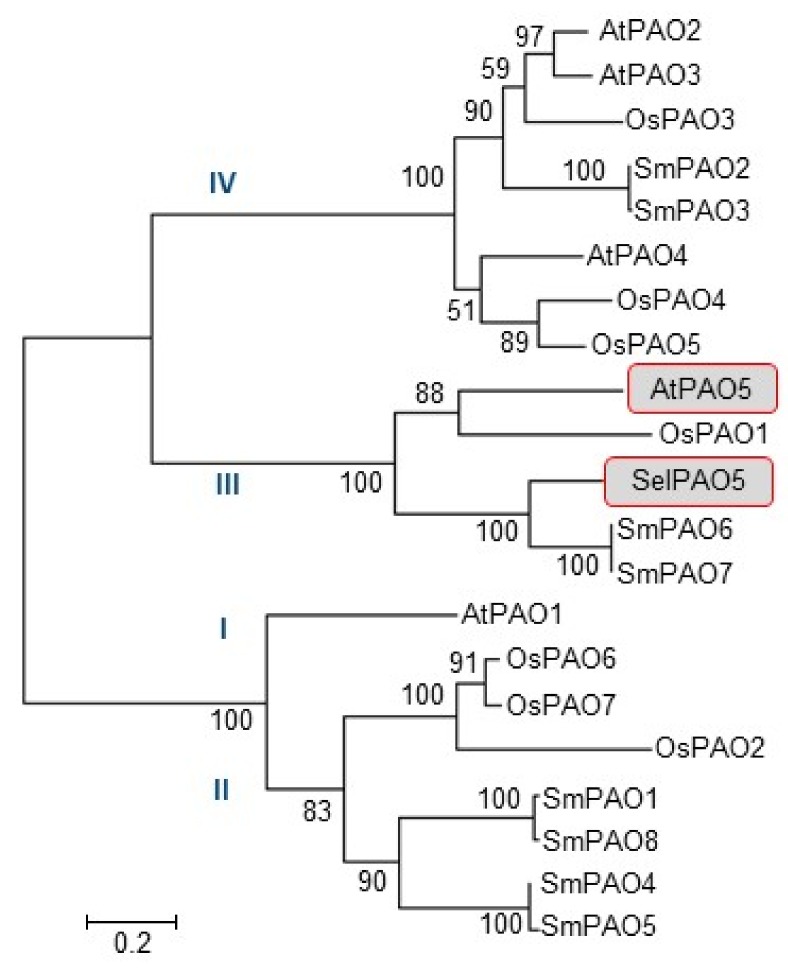
Phylogenetic relationship between SelPAO5, other *Selaginella* PAOs, and selected angiosperm PAOs. The tree was made by alignment of the amino acid sequences using Molecular Evolutionary Genetics Analysis (MEGA 6.0) software [38]. Bootstrap values obtained with 1000 replicates are indicated at the nodes. The genes and accession numbers used are as follows: SelPAO5 (LC036642), SmPAO1 (XP_002965265.1), SmPAO2 (XP_002965599.1), SmPAO3 (XP_002968082.1), SmPAO4 (XP_002969966.1), SmPAO5 (XP_002981437.1), SmPAO6 (XP_002984796.1), SmPAO7 (XP_002985859.1), SmPAO8 (XP_002986593.1), OsPAO1 (NM_001050573), OsPAO2 (NM_001055782), OsPAO3 (NM_001060458), OsPAO4 (NM_001060753), OsPAO5 (NM_001060754), OsPAO6 (NM_001069545), OsPAO7 (NM_001069546), AtPAO1 (NM_121373), AtPAO2 (AF364952), AtPAO3 (AY143905), AtPAO4 (AF364953), AtPAO5 (AK118203).

**Figure 2 plants-08-00099-f002:**
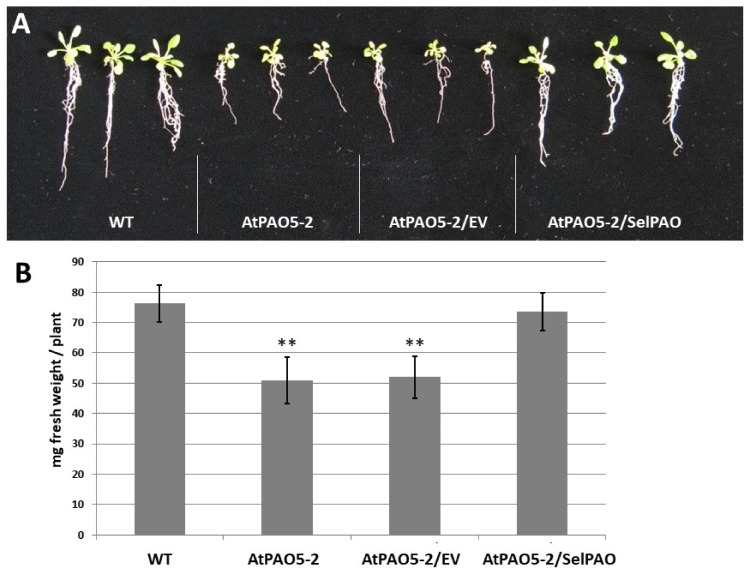
Recovery of thermospermine (T-Spm)-induced growth arrest in *Atpao5-2* by complementation with *SelPAO5*. Wild type plants (WT, Col-0), *Atpao5-2* mutant (AtPAO5-2), *Atpao5-2* transgenic carrying the control empty binary vector pPZP2Ha3(+) [39] (AtPAO5-2/EV), and *Atpao5-2* transgenic line S5#11 carrying the *CaMV35S*-driven *SelPAO5* (AtPAO5-2/SelPAO) were grown vertically for 24 days on half-strength Murashige and Skoog agar medium containing 5 μM T-Spm. Seedlings were carefully picked from the plates and photographed (**A**). The fresh weight of ten seedlings each was determined and the calculated mean including standard deviations displayed in a bar chart (**B**). Asterisks indicate significant differences to fresh weight of WT plants using Student’s *t*-test: ** *p* < 0.01.

**Figure 3 plants-08-00099-f003:**
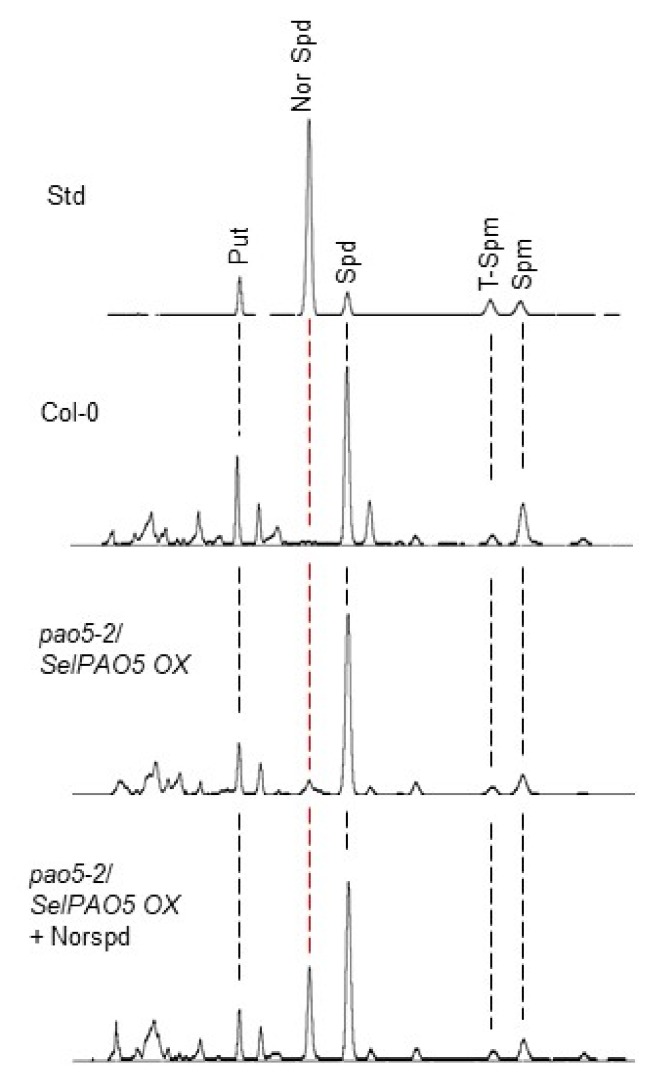
Chromatograms of HPLC analysis of polyamine patterns from *Arabidopsis* wild type plants (Col-0) and *Atpao5-2* mutant expressing *SelPAO5*, respectively. Std = chromatogram of polyamine standards.

**Figure 4 plants-08-00099-f004:**
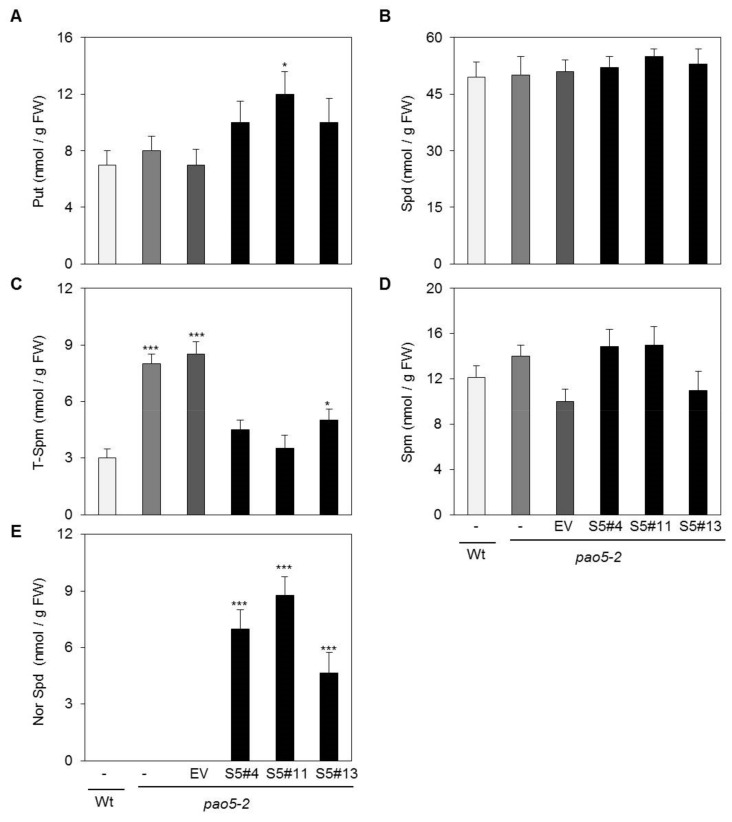
Polyamine content analysis of WT and pao5-2 transgenics under physiological conditions. (**A**) Put; (**B**) Spd; (**C**) T-Spm; (**D**) Spm; (**E**) NorSpd. Plant samples were: WT, *Atpao5-2*, control transgenic EV, and three transgenic lines S5#4, S5#11, and S5#13. *, **, *** indicate significance at a 5%, 1%, and 0.1% level of significance, respectively.

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
