# Peer review of "A Polyamine Oxidase from Selaginella lepidophylla (SelPAO5) can Replace AtPAO5 in Arabidopsis through Converting Thermospermine to Norspermidine instead to Spermidine"

_plants, 2019, doi:10.3390/plants8040099_

Round 1

Reviewer 1 Report

The present MS in well designed and written. 

Small corrections need:

in the abstract in line 22, do not start senetence with also 

in the abstract in line 23, instead of synthesis, maybe conversion is suitable

in the introduction in line 40 instead of conversion, maybe metabolism is suitable

more separated conclusion need in the end of the discussion in order to highlight the novelty.

Author Response

Dear Reviewer,

Thank you very much for your positive assessment of our manuscript.

in the abstract in line 22, do not start sentence with also.

Changed to “ furthermore”

in the abstract in line 23, instead of synthesis, maybe conversion is suitable.

We replaced synthesis by concersion

in the introduction in line 40 instead of conversion, maybe metabolism is suitable. Changed to “ metabolism”.

more separated conclusion need in the end of the discussion in order to highlight the novelty.

In the discussion part we tried to separate the topic of T-Spm homeostasis and appearance of NorSpd in Arabidopsis by SelPAO5 activity.

Reviewer 2 Report

The manuscript by GHM Sagor and collaborators ([Plants] Manuscript ID: plants-453465: "A Polyamine Oxidase from Selaginella lepidophylla (SelPAO5) Can Replace AtPAO5 in Arabidopsis Although Converting Thermospermine to Norspermidine Instead to Spermidine") describes the results of experiments aimed at testing in vivo activity of a Selaginella-derived PAO in Arabidopsis thaliana. This research team previously demonstrated in vitro thermospermine-to-norspermidine back-conversion catalytic activity for this Selaginella enzyme. Therefore, they wanted to test this activity in vivo. To do this, they introduced a Selaginella PAO5 cDNA into an Arabidopsis Atpao5-2 mutant, and verified its ability to rescue a mutant phenotype, defined as "retarded growth" in the presence of thermospermine in the medium. Using this experimental set up, they were able to show that transgenic SelPAO5 can rescue the growth defect of atpao5-2 on thermospermine-containing media. Transgenic plants were also shown to accumulate larger amounts of Put and norspermidine relative to wild type under these conditions. In fact, norspermidine was below detection in wild type Arabidopsis plants as well as mock-transformed atpao5-2 mutant seedlings. The results confirm the importance of thermospermine homeostasis for plant growth and development. It also shows that norspermidine can be produced in Arabidopsis plants expressing the Selaginella PAO5 cDNA. Finally, the authors also used translational fusions of SelPAO5 with GFP to suggest protein localization within the cytoplasm of expressing cells.

This manuscript provides novel information on in vivo function of SelPAO5 in polyamine back-conversion, from thermospermine into norspermidine, a function that was already suggested in a previous manuscript based on in vitro studies. It suggests a possible mechanism for the synthesis of norspermidine in plants. However, I have several concerns about this work, which should be addressed to improve reliability of the conclusions and quality of the publication:

1. The authors use translational fusions between SelPAO5 and GFP to demonstrate cytoplasmic localization of the protein in onion epidermal cells. Unfortunately, these types of experiments are highly sensitive to artifacts related to signal pattern confined to the periphery of the cells (by large vacuoles), a pattern often difficult to distinguish from plasma membrane association without additional plasmolysis experiments. Furthermore, a cytoplasmic-type signal is also often associated with GFP dissociation from the fusion protein. In this case, a nuclear signal is often found, because GFP is small enough to freely diffuse through the nuclear pores. Interestingly, Figure 1B also suggests the existence of a nuclear signal (bottom of the cell), a possibility that is not discussed properly in this manuscript. To be conclusive, these experiments should be accompanied by western blot analyses demonstrating that the fusion protein is not proteolytically cleaved into its main components (free GFP, leading to the cytoplasmic and nuclear signal, and PAO5). Furthermore, to be meaningful, transgenic-rescue experiments should be carried out with the fusion gene, to demonstrate the fusion protein functions properly.

2. One important conclusion of this manuscript is that SelPAO5 can functionally rescue the growth defect associated with Atpao5-2 mutant seedlings growing on thermospermine-containing media. Unfortunately, the results of these experiments are far from being convincing. In fact, wild type, atpao5-2 mutant, and transformed mutant plants expressing the SelPAO5 cDNA, were tested at very high density in petri dishes, without clear boundaries between genotypes on the plates. Pictures were taken from rather far, and they mask the root system. It is very hard to evaluate the true effect that these mutations/transgene, and growth on polyamine containing media, might have on the plants. For instance, looking carefully at Figures 2 B and C, it would appear that at least several S5#13 seedlings grew much slower than some of the other lines (like the wild type or mutant seedlings). In any case, this growth phenotype should be quantified and better interpreted. What does "growth retardation" mean? A more reliable way to address these phenotypes is through the quantification of growth rates, number and size of leaves produced; root growth rate, direction, branching etc.

3. Directly related to point 2, the quantitative aspects of the growth phenotypes described in this manuscript should also be correlated with the level of transgene expression in these plants. The latter should be carefully quantified using RT-qPCR approaches. For instance, is SelPAO5 expressed at lower levels in line S5#13 than in the other two lines? Does level of transgene expression in each line correlate with changes in Put, thermospermine and Norspermidine levels (Figure 4)? (A simple gel electrophoregram showing ampification results from a defined set of PCR cycles is not sufficient for quantification).

4. Considering that the SelPAO5 cDNA is expressed under the control of a 35S promoter (I guess?) in transgenic plants, it would have been helpful to also include a positive control in these experiments, involving transgenic atpao5-2 mutant plants expressing the Arabidopsis AtPAO5 cDNA from the same construct. This experiment would address the possibility that ectopic expression of PAO5 in transgenic Arabidopsis might be responsible for appearance of norspermidine in the samples. Although this outcome is less likely considering the outcome of the in vitro experiments previously published by this group, it would have been useful to eliminate this possibility in vivo as well (which is the whole purpose of this work).

Overall, this is an interesting piece of work, but it requires additional control experiments to ascertain the validity of its conclusions.

Author Response

Dear Reviewer,

Thank you very much for giving suggestions to improve the  manuscript.

1.              The authors use translational fusions between SelPAO5 and GFP to demonstrate cytoplasmic localization of the protein in onion epidermal cells. Unfortunately, these types of experiments are highly sensitive to artifacts related to signal pattern confined to the periphery of the cells (by large vacuoles), a pattern often difficult to distinguish from plasma membrane association without additional plasmolysis experiments. Furthermore, a cytoplasmic-type signal is also often associated with GFP dissociation from the fusion protein. In this case, a nuclear signal is often found, because GFP is small enough to freely diffuse through the nuclear pores. Interestingly, Figure 1B also suggests the existence of a nuclear signal (bottom of the cell), a possibility that is not discussed properly in this manuscript. To be conclusive, these experiments should be accompanied by western blot analyses demonstrating that the fusion protein is not proteolytically cleaved into its main components (free GFP, leading to the cytoplasmic and nuclear signal, and PAO5). Furthermore, to be meaningful, transgenic-rescue experiments should be carried out with the fusion gene, to demonstrate the fusion protein functions properly.

Thank you very much for giving detailed comments on our manuscript.

In a previous paper (Kim et al. Plant Physiology (2014) 165: 1575–1590) we described AtPAO5 and the corresponding mutants. GFP-AtPAO5 fusion constructs were used for cellular localization of AtPAO5 in onion epidermal cells. In the current manuscript such experiments with GFP-SelPAO5 were performed to compare the localization pattern with that of GFP-AtPAO5 of the previous report to make sure that SelPAO5 localization is not different to AtPAO5. In fact the pattern is the same as it was shown for GFP-AtPAO5 and we concluded that the cellular distribution of the transgenically expressed SelPAO5 protein is same as Arabidopsis AtPAO5 protein. For that purpose we think that the shown GFP signal is sufficient to draw this conclusion.

2.              One important conclusion of this manuscript is that SelPAO5 can functionally rescue the growth defect associated with Atpao5-2 mutant seedlings growing on thermospermine-containing media. Unfortunately, the results of these experiments are far from being convincing. In fact, wild type, atpao5-2 mutant, and transformed mutant plants expressing the SelPAO5 cDNA, were tested at very high density in petri dishes, without clear boundaries between genotypes on the plates. Pictures were taken from rather far, and they mask the root system. It is very hard to evaluate the true effect that these mutations/transgene, and growth on polyamine containing media, might have on the plants. For instance, looking carefully at Figures 2 B and C, it would appear that at least several S5#13 seedlings grew much slower than some of the other lines (like the wild type or mutant seedlings). In any case, this growth phenotype should be quantified and better interpreted. What does "growth retardation" mean? A more reliable way to address these phenotypes is through the quantification of growth rates, number and size of leaves produced; root growth rate, direction, branching etc.

We agree that the seedlings were grown in high density in petri dishes and that it is rather problematic that line S5#13 was grown at the far right edge of the plates resulting in an unclear picture. We have already recognized this circumstance and produced another figure with separated seedlings and a quantification of total fresh weight to demonstrate the recovery of growth retardation by expression of SelPAO5 more clearly. Figure and text has been changed accordingly and the previous Fig. 2 was added as supplementary Figure 1.

3.              Directly related to point 2, the quantitative aspects of the growth phenotypes described in this manuscript should also be correlated with the level of transgene expression in these plants. The latter should be carefully quantified using RT-qPCR approaches. For instance, is SelPAO5 expressed at lower levels in line S5#13 than in the other two lines? Does level of transgene expression in each line correlate with changes in Put, thermospermine and Norspermidine levels (Figure 4)? (A simple gel electrophoregram showing ampification results from a defined set of PCR cycles is not sufficient for quantification).

Since the transgene expression is driven by the constitutive 35S promoter we assume that SelPAO5 levels are sufficiently high in all the selected lines when clear amplification of cDNA by PCR is seen. Of course such expression can vary between lines and even between individual plants of one line but for our purpose we only need satisfactory expression to show that SelPAO5 converts T-Spm into NorSpd in planta. Thus, a more detailed analysis of SelPAO5 transgene expression is not necessary at that stage.

4.              Considering that the SelPAO5 cDNA is expressed under the control of a 35S promoter (I guess?) in transgenic plants, it would have been helpful to also include a positive control in these experiments, involving transgenic atpao5-2 mutant plants expressing the Arabidopsis AtPAO5 cDNA from the same construct. This experiment would address the possibility that ectopic expression of PAO5 in transgenic Arabidopsis might be responsible for appearance of norspermidine in the samples. Although this outcome is less likely considering the outcome of the in vitro experiments previously published by this group, it would have been useful to eliminate this possibility in vivo as well (which is the whole purpose of this work).

AtPAO5 did not convert T-Spm or Spm into NorSpd (Kim et al. Plant Physiology (2014) 165: 1575–1590) so we assume that only SelPAO5 activity in the AtPAO5-2 mutant produces NorSpd. Please also see comments to point 3.

Reviewer 3 Report

The manuscript presents new data demonstrating that a homologue of AtPAO5 (polyamine oxidases in Arabidopsis thaliana) from the lycophyte Selaginella lepidophylla (SelPAO5) is able to back-convert T-Spm to the uncommon polyamine norspermidine (NorSpd) instead of Spd in the atpao5-2 transformed plants. Investigation is based on a variety of physiological (phenotyping), biochemical and molecular parameters. The authors concluded that a balanced T-Spm homeostasis is necessary for normal plant growth.

The experimental design and analytical procedures are well described. Sections ‘Results’ and ‘Discussion’ are well written and informative. The results are well described and evaluated statistically. Further the results are well discussed and the arguments in the ‘Discussion’ section are sound.

In conclusion, the manuscript falls within the scope of the journal, it is definitely well written with good English style and it presents novel findings with high scientific value. Therefore, I recommend the manuscript to be accepted for publication.

Author Response

Dear Reviewer,

Thank you very much for your positive assessment of our manuscript.

We have changed several parts in our manuscript for improvement.

Round 2

Reviewer 2 Report

This revised version of the manuscript by Sagor et al. addresses some, but not all, the concerns that were raised on the first draft. The concern initially raised about the qualitative assessment of a rescue of TSpm-induced Atpao5-2 growth arrest by transgenic SelPAO5 has been addressed by including new fresh-weight measurements of seedling sizes for the various genotypes under consideration. The only additional request I would have regarding Figure 1b would be to include a statistical analysis of differences between genotypes.

Another important concern I had about this manuscript related to the protein localization experiment involving transient expression of a reporter fusion into onion epidermal cells, which the authors interpreted as indicating cytoplasmic association. However, the fluorescence patterns associated with a control GFP transgene and the fusion protein indicated similar patterns, at the periphery of transformed onion epidermal cells. In addition, there was evidence of signal association with the nucleus. This type of signal is, unfortunately, very often an artifact due to the GFP moiety being dissociated from the fusion protein (or the fusion being incompletely translated), and diffusing through the cytoplasm as well as into the nucleus due to its small size. As a consequence, most researchers observing a similar signal in this type of assay would perform a plasmolysis experiment (treating the cells with high osmoticum to reduce the size of the vacuole, thereby allowing to distinguish signal associated with the plasma membrane from those associated with the cytoskeleton), and also perform a western blot analysis of transformed material to verify that no free GFP can be observed in transgenic cells (which would suggest its proteolytic cleavage from the fusion protein, or incomplete translation in the case of an amino-terminal fusion).

In response to my request for such control experiments, the authors simply reply that the pattern they observe with this GFP-SelPAO5 protein in transformed onion epidermal cells is similar to the one displayed by similar GFP-AtPAO5 fusions in previous experiments published in Ahou et al. (2014) (using different cell types), and that this is enough to conclude cytoplasmic association. This is not very satisfactory to me because almost any protein that is not associated with some of the organelles or the vacuole will display a similar pattern of localization in onion epidermal cells.  In this manuscript, the similarity in localization patterns displayed by At- and SelPAO5 proteins is one of two parameters used to suggest that these proteins are orthologous (lines  164-165). Yet, the authors conclude from the rest of their analyses that the two proteins function differently (they catalyze the production of different compounds). Therefore, it does not seem logical to assume that similarity in incompletely controlled localization patterns is sufficient to confirm a localization of SelPAO5 from the data displayed in Figure 1B (this is circular argumentation, in fact!).

I think there is a definite need to include controls in the protein localization studies, involving plasmolyzed cells (to eliminate the possibility of association with the plasma membrane), and another control that unambiguously addresses possible association with the nucleus (a Western blot, for instance, could address the possibility that some GFP is cleaved off the fusion protein, or the fusion was incompletely translated due to poor codon usage, and the resulting free GFP traverses the nuclear pores to generate a fluorescence signal within the nucleus).

In addition to the problem I have with the lack of controls included in the protein localization experiment, I would also like to suggest inclusion of some description of the experimental set up in the legend to Figure 1, which lacks information on panel b.

The third point I raised regarding the need to include a more quantitative RT-qPCR analysis of expression levels between transgenic plants displaying distinct levels of phenotypic rescue to correlate expression with degree of rescue (and amount of NorSpd produced) seems to be discounted by the authors, who conclude that it is sufficient to show that the gene is expressed in lines that actually produce \ NorSpd compared to those that do not. While I do understand their reasoning, I also realize that many things can happen when over-expressing an enzyme in a different context. For instance, I still believe it would be very useful to include in this experiment a control that over-expresses the Arabidopsis AtPAO5 protein from the same construct, and quantitatively compare expression of all transgenic lines as well as NorSpd production. The authors argue that they have already demonstrated that AtPAO5 does not produce NorSpd in plants (Kim et al., 2014). However, a quick look at this paper shows that the transgenic plants used in the cited paper involved the AtPAO5 genomic DNA as a transgene, rather than a fusion to the strong CaMV 35S promoter. Furthermore, I do not believe that the cited paper reported on NorSpd production by these plants.

Finally, I would like to once again suggest replacing the term 'retarded' ('retarded growth'; 'retarded phenotype') by 'reduced'. Indeed, 'retarded' is somewhat associated with 'delayed'. Unless it can be shown that these phenotypes eventually catch up with wild-type plants, I think 'reduced' (or other term similar to it) would be more appropriate here.

Overall, I do not think that the authors have adequately addressed the comments raised on the first draft of this manuscript.

Author Response

Dear Reviewer 2, thank you very much for your detailed report on our manuscript.

Concerning our data of cellular localization of GFP-SelPAO5 in onion epidermal cells you requested a more detailed analysis by e.g. immunoblot experiments. However, at this stage we cannot perform such experiments in the given time frame. Alternatively we erased such data from our manuscript because actually they do not touch the main message, SelPAO5 can replace AtPAO5 function although producing NorSpd.

Furthermore, the expression analyses of SelPAO5 in three different transgenic Arabidopsis lines compared to AtPAO5 gene was repeated in a qRT-PCR approach replacing the RT-PCR result that was displayed as gel electrophoresis pattern (Supplementary Figure S1).

We also replaced “growth retarded” by “growth reduced” throughout the manuscript as you suggested.

We hope with these modifications the content of our manuscript acceptably demonstrates that expression of T-Spm specific PAO of S. lepidophylla (SelPAO5) can replace AtPAO5 function by curing the growth repression in the AtPAO5-5 mutant although producing Nor-Spermidine instead of spermidine.

Changes within the manuscript are highlighted by the Track Changes function of MS-Word.

Round 3

Reviewer 2 Report

In this revised version of the manuscript the authors adequately address the comments and suggestions I made on the previous draft. They removed the protein localization data because they do not bring much to the conclusion, while being potentially misleading for lack of adequate control. They also included in this revision the results of a qRT-PCR analysis of transgene and endogene levels, although they did not attempt to correlate relative expression levels between transgenic lines and TSpm levels, where differences seem relevant between line S5#13 vs #4 and #11 (lower transgene expression associated with higher Tspm level and lower NorSpd level in line Sr#13). An ANOVA analysis would have helped elucidate the statistical significance of these differences. Finally, they replaced the terminology 'retarded growth' by 'reduced growth' in most sections of the manuscript. Overall, this new draft is improved. Minor editorial suggestions follow:

'growth reduced phenotype' should probably be replaced by 'reduced growth phenotype' at several sections within the manuscript. Similarly, the terminology 'retarded growth' is still used predominantly in the Discussion, where it should also be replaced by 'reduced growth'.

The accepted terminology for genes, alleles and protein names in Arabidopsis thaliana should be followed as well. Mutant alleles should be in lower case, italics. Gene names and wild type alleles should be in capital letters, italics. Protein names should be written in capital letters, normal font.

In Materials and Methods, the sentence written in lines 202-203 does not make much sense to me. Are the agar plates vertical, or inclined at an angle of 75 degrees? What does the angle of 75 degrees relate to? (What is the reference point here; is it gravity?).

Author Response

Dear Reviewer, again thank you very much for handling our manuscript.

We changed 'growth reduced phenotype' by 'reduced growth phenotype' and 'retarded growth' by 'reduced growth' throughout the manuscript.

The terminology for gene names, proteins and mutants has been corrected accordingly.

The unclear sentence describing the growth of Arabidopsis plants on agar plates was changed:

Agar plates were kept upright under the angle of 75º to ground to allow gravity-induced plant growth on the agar surface.

Plants EISSN 2223-7747 Published by MDPI AG, Basel, Switzerland RSS E-Mail Table of Contents Alert
Back to Top